# Diagnostic Properties of Different Serological Methods for Syphilis Testing in Brazil

**DOI:** 10.3390/diagnostics15121448

**Published:** 2025-06-06

**Authors:** Suelen Basgalupp, Thayane Dornelles, Luana Pedrotti, Aniúsca dos Santos, Cáren de Oliveira, Giovana dos Santos, Emerson de Brito, Ben Hur Pinheiro, Ana Cláudia Philippus, Álisson Bigolin, Pamela Cristina Gaspar, Flávia Moreno, Gerson Pereira, Maiko Luis Tonini, Eliana Wendland

**Affiliations:** 1Hospital Moinhos de Vento, PROADI-SUS, Porto Alegre 90560-030, Brazil; suelenbasgalupp@gmail.com (S.B.); thayanemdornelles@gmail.com (T.D.); lu.pedrotti75@gmail.com (L.P.); aniusca.vieira@gmail.com (A.d.S.); caren.ndo@gmail.com (C.d.O.); santos.giovanat@gmail.com (G.d.S.); benhurgraboski@gmail.com (B.H.P.); 2Department of Community Health, Federal University of Health Science of Porto Alegre, Porto Alegre 90050-170, Brazil; emersonb@ufcspa.edu.br; 3Department of HIV/Aids, Tuberculosis, Viral Hepatitis and Sexually Transmitted Infections, Health and Environmental Surveillance Secretariat, Brazilian Ministry of Health, Brasília 70058-900, Brazil; ana.philippus@aids.gov.br (A.C.P.); alisson.bigolin@aids.gov.br (Á.B.); pamela.gaspar@aids.gov.br (P.C.G.); flavia.moreno@aids.gov.br (F.M.); gerson.pereira@aids.gov.br (G.P.); maiko.tonini@saude.gov.br (M.L.T.)

**Keywords:** syphilis diagnosis, serological tests, POC, VDRL, RPR, ELISA, TPHA

## Abstract

**Background/Objectives**: Syphilis remains a significant public health challenge worldwide. Accurate and efficient diagnostic tools are essential to controlling the spread of the disease. Current diagnostic approaches primarily rely on serologic treponemal tests (TTs) and nontreponemal tests (NTTs). The aim of this study was to evaluate the diagnostic properties of various serological methods for syphilis diagnosis. **Methods**: Samples were collected from participants of the Health, Information, and Sexually Transmitted Infection Monitoring (SIM study) between March 2020 and May 2023, using convenience sampling at a mobile health unit in Porto Alegre, Brazil. A total of 250 individuals were tested using the point-of-care (POC) lateral flow treponemal test, Venereal Disease Research Laboratory (VDRL) test, Rapid Plasma Reagin (RPR) test, Enzyme-Linked Immunosorbent Assay (ELISA), and *Treponema pallidum* hemagglutination assay (TPHA). Of these, 125 participants tested positive for syphilis in the POC screening. Diagnostic properties such as sensitivity, specificity, and predictive values were assessed for the POC test, ELISA, and VDRL test. The TPHA was used as the reference standard for the TT, and the RPR test as the reference standard for the NTT. **Results**: Among individuals with positive POC test results, 97.6% (122/125) were also positive by the ELISA, and 85.6% (107/125) were positive by the TPHA. Additionally, 48.0% (60/125) and 42.4% (53/125) tested positive by the VDRL and RPR tests, respectively. Using the TPHA as a reference, TT tests showed sensitivities of 97–98% and specificities of 93–95% for detecting anti-*Treponema pallidum* antibodies using the ELISA and POC test, respectively. For the NTT, the VDRL test demonstrated a sensitivity of 98% and a specificity of 95% compared to the RPR test. The kappa coefficients were 0.85 for the POC test vs. the TPHA, 0.81 for the ELISA vs. the TPHA, and 0.89 for the VDRL vs. the RPR tests, indicating substantial agreement. **Conclusions**: This study highlights a good diagnostic performance and high agreement levels among the evaluated serological tests for syphilis, reinforcing their utility in clinical and public health settings, as well as epidemiological studies.

## 1. Introduction

Syphilis is a sexually transmitted infection (STI) caused by the bacterium *Treponema pallidum* (*T. pallidum*). Transmission occurs primarily through direct contact with lesions during oral, vaginal, or anal sexual activity. Additionally, syphilis can be transmitted vertically from mother to fetus during pregnancy or childbirth [1].

The global prevalence of syphilis remains a significant public health concern, mainly in resource-limited countries with inadequate investments in primary healthcare. According to the World Health Organization (WHO), an estimated 7.1 million new cases of syphilis occurred worldwide in 2022, with the highest burden occurring in low- and middle-income countries [2]. Among the most pressing challenges are the high morbidity and mortality associated with congenital syphilis, which continue to represent a major public health issue globally [1]. In Brazil, syphilis persists as a critical health concern, with a rising number of reported cases in recent years. In 2023, the country recorded 242,826 cases of acquired syphilis, corresponding to a detection rate of 113.8 cases per 100,000 inhabitants. Monitoring the prevalence of and trends in syphilis is essential for informing and implementing effective prevention and elimination strategies [3]. Serological testing is the most employed laboratory technique for diagnosing syphilis [4]. Various methods are available, broadly categorized into treponemal and nontreponemal tests. Treponemal tests (TTs), such as the *Treponema pallidum* particle hemagglutination assay (TPHA), point-of-care (POC) immunochromatographic/lateral flow treponemal tests, and enzyme immunoassays (EIAs), target antibodies to *T. pallidum* antigens and exhibit high specificity. On the other hand, nontreponemal tests (NTTs), such as the Venereal Disease Research Laboratory (VDRL) and Rapid Plasma Reagin (RPR) tests, detect antibodies that react with non-specific antigens released during infection. The combined use of TTs and NTTs constitutes a comprehensive diagnostic approach, enabling the accurate screening, detection, staging, and monitoring of syphilis [5]. The Technical Manual for the Diagnosis of Syphilis, published by the Brazilian Ministry of Health, recommends a testing algorithm that integrates TTs and NTTs to increase the positive predictive value and help to differentiate between current and past infection [5].

The performance of different serological tests varies throughout the stages of syphilis. In the early stage (primary syphilis), a TT is generally more sensitive than an NTT. Relying on nontreponemal tests during early infection may result in misdiagnosis and delays in treatment. As the disease progresses to the secondary stage, the sensitivity of nontreponemal tests increases and may approach 100% [6].

Diagnostic challenges may also arise depending on the testing algorithm employed, as some laboratories prioritize specific tests as primary diagnostic tools, particularly when clinical symptoms are unclear, as is often the case with syphilis. Importantly, test results reflect an individual’s serological status and should be interpreted in conjunction with their clinical and epidemiological history [5].

The aim of this study was to evaluate the diagnostic properties of different serological methods for syphilis diagnosis and to assess the different diagnostic flowcharts used in Brazil.

## 2. Materials and Methods

### 2.1. Study Population and Sample Collection

This was a cross-sectional study that evaluated individuals with and without syphilis infection based on the results of a POC test and who were enrolled in the SIM study. The research was conducted using a mobile health unit stationed in high-traffic areas of Porto Alegre, Rio Grande do Sul, Brazil. A convenience sample of 250 whole-blood (for the POC test) and serum samples was collected between March 2020 and May 2023. The study protocol has been published previously (NTC04753125, trial registration) [7]. 

Briefly, people over 18 years old answered a short questionnaire collecting sociodemographic information (sex, age, education, ethnicity, marital status, income, employment status, place of residence, and nationality) before the fingertip sample collection for the POC test for syphilis, HIV, HBV, and HCV. All participants who tested positive for syphilis on the POC test were invited to provide a blood sample via venipuncture, collected in a BD Vacutainer^®^ SST^®^ II Advance^®^ tube with separating gel, for serological testing. Samples were centrifuged at 2000 rcf for 10 min to obtain serums. We used a random sample from a large prevalence study (SIM study), along with a consecutive sample of participants who tested negative for all STIs evaluated by POC testing, to minimize potential cross-reactivity.

The study adhered to the principles outlined in the Declaration of Helsinki and was approved by the Institutional Research Ethics Board (REB) (protocol number 26185219.0.0000.5330). Written informed consent was obtained from all participants after they received detailed information about the study objectives and procedures.

### 2.2. Sample Size Calculation

A sample size of 250 subjects was calculated to estimate the sensitivity and specificity of diagnostic tests for syphilis with a 15% amplitude for the confidence interval and a 95% confidence level, assuming half of the sample tested positive and half tested negative, according to the gold-standard test defined. The calculation was performed for each reference study and the highest result was adopted as the necessary sample size [8,9,10]. This calculation was carried out using the PSS Health tool, online version [11]. Five serological tests were performed and their characteristics are shown below (Table 1).

### 2.3. Treponemal Tests

#### 2.3.1. The Point-of-Care (POC) Lateral Flow Treponemal Test 

The detection of total *anti-Treponema pallidum* antibodies (IgG, IgM, and IgA) was assessed using a lateral flow point-of-care test. This qualitative test detects a single band for antibodies without distinguishing among them. The POC SÍFILIS BIO (Bioclin, Belo Horizonte, Brazil) was employed using capillary-whole-blood samples. Testing was performed by trained nurses, according to the manufacturer’s instructions. A 10 µL blood sample collected via finger prick was used, discarding the first drop. The test results were read from 15 to 20 min after the addition of the diluent. Tests showing visible bands in both the test (T) and control (C) areas of the cassette were considered to be positive. Tests where the C band was absent were deemed invalid and repeated using a new cassette. According to the manufacturer, the C region contains a polyclonal anti-*T. pallidum* antibody, and the T region contains a recombinant *T. pallidum* antigen immobilized on the membrane. Although the specific recombinant antigen is not disclosed, it is designed to bind to treponemal antibodies present in the sample. The manufacturer reports a clinical sensitivity greater than 99.9% and a specificity of 99.8% for the detection of treponemal antibodies.

#### 2.3.2. Enzyme-Linked Immunosorbent Assay (ELISA)

The BIOLISA SÍFILIS Ac TOTAL kit (Bioclin, Belo Horizonte, Brazil; Cat # K240) is an immunoenzymatic solid-phase assay detecting IgG, IgM, and IgA antibodies to *T. pallidum* in human serum. Samples were diluted and added to antigen-coated wells, and incubated and washed. Positive samples displayed specific antibody binding. Enzyme-conjugated antigens were added, binding to the antibodies. Substrate solution incubation produced a blue color indicating antibody presence, with absorbance measured at 450 nm/630 nm using a microplate reader (Multiskan™ FC Microplate Photometer—Thermo Scientific™, Waltham, MA, USA. Results were semiquantitatively evaluated, with a ratio of <0.9 considered negative, ≥0.9 to ≤1.1 considered undetermined, and >1.1 considered positive, following the manufacturer’s instructions. Blank absorbance subtraction was applied to all samples. The manufacturer reports a clinical sensitivity greater than 99.9% and a clinical specificity also exceeding 99.9% for this assay. The TPHA was used as the reference standard to evaluate the ELISA’s performance, consistent with current diagnostic protocols for treponemal testing.

#### 2.3.3. The Treponema Pallidum Hemagglutination Assay (TPHA)

The TPHA test uses preserved avian erythrocytes coated with antigens of *T. pallidum* to bind with specific antibodies present in patient serum. The cells were suspended in a diluent containing components to eliminate non-specific reactions. Positive reactions were characterized by hemagglutination. The ASI TPHA test (Arlington Scientific, Springville, UTAH, USA; Cat # 980200E) was analyzed in this study and conducted according to the manufacturer’s instructions. Briefly, 10 µL of serum samples were diluted into 190 µL of TPHA diluent and added to each of the 2 wells. Test cells and control cells were gently mixed to ensure thorough resuspension, and 75 µL of the test cells were added to the 1st well, while 75 µL of the control cells were added to the 2nd well of the plate. After mixing, the plate was incubated at room temperature on a vibration-free surface for 45 min. The interpretation of results was performed by the same observer for all samples. The TPHA results were adopted as the gold standard.

### 2.4. Nontreponemal Tests

#### 2.4.1. The Venereal Disease Research Laboratory (VDRL) Test

This is a nontreponemal test for the detection of reagin/cardiolipin antibodies in human serum by flocculation, which is observed using a microscope and the result interpretation is dependent upon the observer. All results were interpreted by the same observer. When present in individuals infected with *T. pallidum*, the reagins are detected in the serum by reaction with a purified and stabilized cardiolipin antigen producing a flocculation that is visible under the microscope. The VDRL (Wiener lab, Rosario, Argentina) kit was used in this study according to the manufacturer’s instructions. Briefly, a serial dilution was performed at a ratio of 1:2 from the pure sample to a dilution of 1:32. In the case of the reagent being present up to a 1:32 dilution, a new test was performed up to a 1:1024 dilution. The results were obtained from the last dilution in which flocculation occurred.

#### 2.4.2. The Rapid Plasma Reagin (RPR) Test

This is a nontreponemal test for the detection of reagin/cardiolipin antibodies in human serum by flocculation. The RPR Corado (Laborclin Produtos para Laboratórios, Vargem Grande, Pinhais/PR Paraná, Brazil) kit was used in this study. The sample was added to an antigenic suspension in the reading card and this mixture was submitted through an orbital rotation process for 8 min. During this time, cholesterol particles coated with cardiolipin and lecithin will flocculate if the samples contain reagins. This flocculation reacts with the toluidine red particles forming clumps that are visible to the naked eye. As result interpretation was observer-dependent, all results in this study were evaluated by the same observer to ensure consistency. The titles of positive samples were obtained by serial dilutions at a ratio of 1:2 from the pure sample to the dilution of 1:32. If the sample was reagent up to a 1:32 dilution, a new test was performed up to a 1:1024 dilution. The results were obtained from the last dilution in which flocculation occurred. The RPR test was adopted as the gold standard for NTTs.

### 2.5. Statistical Analysis

Descriptive statistics were used to summarize the variables of interest. The categorical variables were expressed as absolute and relative frequencies. Comparisons between variables were made using Chi-squared or Fisher’s Exact tests, when appropriate. For these comparisons, a *p*-value of less than 0.05 was considered to be statistically significant.

Sensitivity, specificity, positive predictive value (PPV0, negative predictive value (NPV), and their respective 95% confidence intervals (CIs) were calculated, as well as the agreement between different serological treponemal and nontreponemal tests.

Agreement between different serological tests was evaluated using Cohen’s kappa score. Cohen’s kappa was classified as follows: 0.00–0.20, slight; 0.21–0.40, fair; 0.41–0.60, moderate; 0.61–0.80, substantial; 0.81–1.00, almost perfect [12]. Statistical analysis was carried out using R software version 4.1.3 [13].

## 3. Results

Of the 250 individuals recruited for this study, 125 tested positive and 125 tested negative for syphilis in the POC screening test. The majority of participants were under 45 years old, identified as white, and had either completed or not completed higher education. There were no differences in sex or age ranges between the POC-positive and -negative test groups. However, the POC-positive group had a higher frequency of Black and Brown individuals, as well as a lower educational level (Table 2).

Overall, all tests presented a high sensitivity and specificity, above 93%. The NPV is above 98%, ensuring that the test was a good diagnostic test, with very low false-negative results. The PPV of the VDRL test compared to the RPR test is slightly lower, presenting a high percentage of false-positive tests (PPV = 85%), but with an excellent performance as a diagnostic test (NPV = 99%) (Table 3) [14]. The Kappa agreement was considered to be high for all comparisons (>0.80).

In relation to the comparison between the POC test and TPHA, 11 participants were excluded from the total number due to inconclusive results in the TPHA test. In the same way, 13 participants were excluded from the total *n* in the comparison between the TPHA and ELISA.

Comparing the titers of nontreponemal tests, the agreement between the VDRL and RPR tests was weak. In general, the RPR test results showed a reduction of one titer compared to the VDRL test results (e.g., from 1:16 to 1:8), especially for results above 1:8 titers (Figure 1).

Several diagnostic algorithms for syphilis diagnosis are described. In this study, four flowcharts (A→D) addressing the reverse algorithm were generated, according to Flowchart 2 from the Technical Manual for the Diagnosis of Syphilis from the Brazilian Ministry of Health, which recommends that the laboratory diagnosis begins with a POC treponemal test, followed by a nontreponemal test, and a third treponemal test using a different methodology from the first in cases of discordance in the first two tests [5,15]. Considering the A to D flowcharts, the sensitivity and specificity were 100% and 95% for A and C, respectively, and 100% and 98% for B and D, respectively (Figure 2).

Of the 60 individuals who tested reactive in the VDRL test, 26 (43.3%) had titers of 1:1 (14/26) or 1:2 (12/26). Among these, four reported having already completed the treatment. The remaining 34 individuals who tested reactive in the VDRL test presented the following titers: 1:4 titer (*n* = 7), 1:8 titer (*n* = 7), 1:16 titer (*n* = 7), 1:32 titer (*n* = 6), 1:64 titer (*n* = 6), 1:128 titer (*n* = 7), and 1:256 titer (*n* = 1).

## 4. Discussion

This study demonstrated high agreement and good performance in the diagnostic properties of the tests used in comparison to the established gold standard [14]. Among the four syphilis diagnostic algorithms presented, the ELISA exhibited the best performance, with the higher detection of individuals with syphilis compared to the TPHA, which is considered to be the gold standard in most studies [10,16,17]. The ELISA has the potential to become the preferred test for confirming syphilis diagnoses due to several advantages, including automated result interpretation and independence from observer variability, which minimizes inconsistencies across laboratories. In this study, two gold-standard tests were employed, one for each type of test (a TT and an NTT), unlike other studies that typically use the TPHA as the gold standard for all tests [10,16,17]. Regarding the positivity rate, the TT demonstrated a higher detection rate of reactive individuals compared to the NTT, which is consistent with other studies [18,19,20].

While all flowcharts presented comparable results, demonstrating high sensitivity and specificity, the VDRL test alone exhibited higher rates of positivity when compared to the RPR test. Furthermore, the ELISA is recognized for its enhanced sensitivity and specificity for syphilis in contrast to nontreponemal tests (e.g., the VDRL test). This observation is consistent with the results reported by other researchers [5]. Therefore, incorporating the VDRL test can potentially eliminate the requirement for a third diagnostic test, and the use of an ELISA can mitigate the occurrence of inconclusive findings.

The traditional syphilis screening algorithm begins with a nontreponemal test. If the result is positive, a treponemal test is performed [21,22]. However, many clinical laboratories have used a reverse algorithm for syphilis screening [23,24,25]. The reverse algorithm starts with a treponemal test, followed by a nontreponemal test if the results are positive. In the case of discordant results, a second treponemal test is performed [22,24,26]. By employing the reverse algorithm, we can enhance the sensitivity in diagnosing primary syphilis, as the NTT requires more time to detect recent infections. Similar findings have been obtained in other studies [25,27].

An important aspect to be considered is that the stages of the disease were not determined, which can influence the diagnostic properties of the tests, as these can vary depending on the clinical stage of the disease and the test used. Due to insufficient clinical information about the participants’ symptoms, disease stage, and previous treatment status, we were unable to distinguish between active syphilis and a serological scar [28]. This lack of detailed clinical data constitutes an important limitation of our study, as it restricts the ability to fully assess the diagnostic accuracy across different disease stages. Another important point to highlight is that the study was conducted in a subsample from the southern region of Brazil, which limits the generalizability of the results to other populations.

It should be emphasized that several medical conditions can lead to false positive results in NTTs, such as autoimmune disorders (including systemic lupus erythematosus), human immunodeficiency virus (HIV) infection, cancers, pregnancy, Lyme disease, lymphoma, rheumatoid arthritis, and old age, among others [8,29,30,31,32,33,34]. This is partly because NTTs detect antibodies against non-specific antigens, such as cardiolipin, lecithin, and cholesterol, which can also be produced in response to various non-syphilitic conditions. To address this limitation, reactive NTT results should always be confirmed with a specific TT, especially in populations with a high prevalence of conditions associated with false positives. Additionally, clinical correlation and, when available, follow-up testing can help distinguish between false positives and true active infections.

Furthermore, it is important to highlight that there is variability in reading patterns, whether between different observers or due to changes in the test brand used. Our study consistently utilized the same brand for all conducted tests, and the interpretation of results was performed by the same researcher on all analyzed samples. There is no described pattern in the literature regarding the variation in titers among different nontreponemal tests. However, our study observed a reduction in titers from the VDRL test compared to the RPR test for the same sample, especially in cases of high titers. For example, a participant showed a result of 1:64 for the VDRL test, while for the RPR test, the same participant had a result of 1:32.

Chemiluminescence is rarely used for syphilis diagnosis in Brazil due to its higher cost compared to traditional tests, the need for advanced laboratory infrastructure that is not widely available in remote or underserved areas, and the continued reliance on well-established, cost-effective methods like the VDRL test, FTA-ABS test, and TPHA, which are considered to be efficient for large-scale screening [18,35,36].

## 5. Conclusions

The ongoing challenges in diagnosing syphilis, particularly against a backdrop of rising global prevalence, underscore the need to explore more-effective diagnostic strategies. This study analyzed various approaches to using diagnostic flowcharts using the reverse algorithm for syphilis. All comparisons demonstrated satisfactory performance, showing high concordance in both treponemal and nontreponemal tests. The findings suggest that the ELISA could be a promising alternative for confirming syphilis diagnoses due to its automated result interpretation and observer independence, thus reducing result variability across laboratories.

However, it is important to note that all serological tests exhibit variable rates of false-positive and false-negative results. Therefore, results should be interpreted within a comprehensive clinical and epidemiological context. Future studies should aim to correlate clinical findings with serologic outcomes to further improve diagnostic accuracy.

## Figures and Tables

**Figure 1 diagnostics-15-01448-f001:**
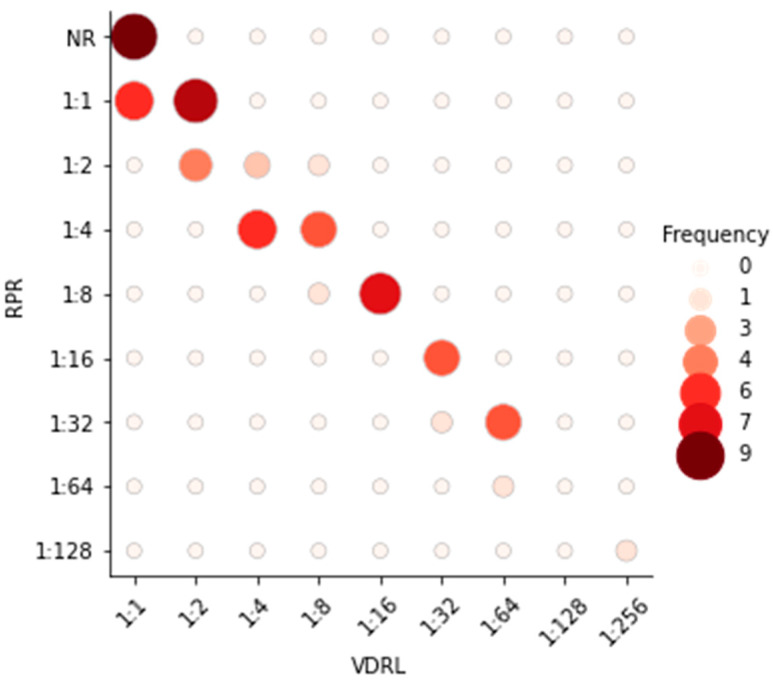
Comparison between nontreponemal test titers (VDRL vs. RPR tests). On the *x*-axis are the titers of the VDRL test, while the *y*-axis corresponds to the titers of the RPR test. The circles represent the results identified in both tests, with the size of the circle being directly proportional to the frequency of the results found. Additionally, the intensity of the colors represented in the circles is directly proportional to the frequency of the identified results.

**Figure 2 diagnostics-15-01448-f002:**
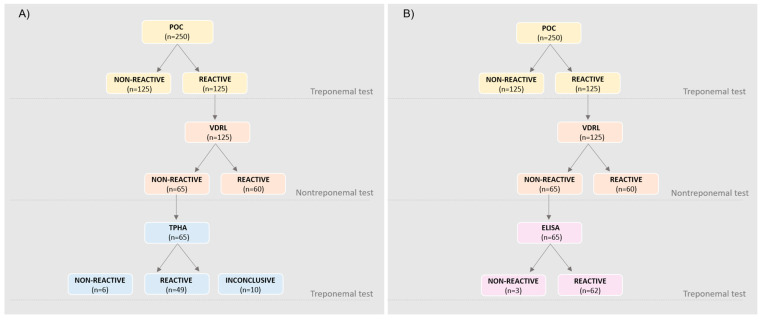
Syphilis positive testing flowchart with reverse algorithm showing POC-positive tests following treponemal and nontreponemal testing. (**A**) POC test followed by VDRL test and TPHA; (**B**) POC test followed by VDRL test and ELISA; (**C**) POC test followed by RPR test and TPHA; (**D**) POC test followed by RPR test and ELISA.

**Table 1 diagnostics-15-01448-t001:** Characteristics of the diagnostic tests used.

Characteristics	Treponemal Test			Nontreponemal Test	
	POC Test	ELISA	TPHA	VDRL Test	RPR Test
Method	Immunochromatographic	Immunoassay	Hemagglutination	Agglutination	Agglutination
Specimen type	Whole blood	Serum	Serum	Serum	Serum
Specimen volume required	10 µL	5 µL	10 µL	100 µL *	100 µL *
Dilution	NA	Manual dilution	Manual dilution	Manual dilution	Manual dilution
Time to test result **	20 min	110 min	90 min	10 min	14 min
Interpretation	Dependentobserver	Automated	Dependent observer	Dependent observer	Dependent observer
Manufacturer	Bioclin	Bioclin	ASI	Wiener	Laborclin

* 100 µL was considered to be a diluted sample. ** The time to test result was considered to be the estimated time to perform the test.

**Table 2 diagnostics-15-01448-t002:** Sociodemographic characteristics of included participants.

Characteristics	Total N (%)	POC+ N (%)	POC− N (%)	*p*-Value
Sex				
Male	123 (49.2)	67 (54.47)	56 (45.53)	0.206
Female	127 (50.8)	58 (45.67)	69 (54.33)	
Age (years)				
18–29	83 (33.2)	36 (43.37)	47 (56.63)	0.282
30–45	88 (35.2)	46 (52.27)	42 (47.73)	
46–59	40 (16.0)	19 (47.50)	21 (52.50)	
60+	39 (15.6)	24 (61.54)	15 (38.46)	
Ethnicity				
White	148 (60.16)	64 (43.24)	84 (56.76)	0.036
Black	43 (17.48)	26 (60.47)	17 (39.53)	
Brown	52 (21.14)	29 (55.77)	23 (44.23)	
Others	3 (1.22)	3 (100.00)	0 (0.00)	
Education Level				
Elementary	65 (26.0)	46 (70.77)	19 (29.23)	<0.001
Secondary	90 (36.0)	43 (47.78)	47 (52.22)	
University	95 (38.0)	36 (37.89)	59 (62.11)	

**Table 3 diagnostics-15-01448-t003:** Diagnostic properties of POC test and ELISA using TPHA as a gold standard, and VDRL test using RPR test as a gold standard.

	True Positive (*n*)	True Negative (*n*)	False Positive (*n*)	False Negative (*n*)	Sensitivity % (95% CI)	Specificity % (95% CI)	PPV % (95% CI)	NPV % (95% CI)	KAPPA
POC test vs. TPHA	107	123	7	2	98.0 (94.0–100.0)	95.0 (89.0–98.0)	94.0 (88.0–97.0)	98.0 (94.0–100.0)	0.85
ELISA vs. TPHA	106	119	9	3	97.0 (92.0–99.0)	93.0 (87.0- 97.0)	92.0 (86.0–96.0)	98.0 (93.0–99.0)	0.81
VDRL test vs. RPR test	53	187	9	1	98.0 (90.0–100.0)	95.0 (91.0–98.0)	85.0 (74.0–93.0)	99.0 (97.0–100.0)	0.89

POC: Point-of-care test; positive and negative groups for syphilis screened by POC test, based on lateral flow point-of-care test using fingerstick whole-blood samples. VDRL: Venereal Disease Research Laboratory; positive and negative groups for syphilis tested by VDRL test using serum samples. RPR: Rapid Plasma Reagin; positive and negative groups for syphilis tested by RPR test using serum samples. ELISA: Enzyme-Linked Immunosorbent Assay; positive and negative groups for syphilis detected by ELISA using serum samples. TPHA: *Treponema pallidum* hemagglutination assay; positive and negative groups for syphilis detected by TPHA using serum samples.

## Data Availability

The data that support the findings of this study are available on request from the corresponding author.

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
