# Peer review of "Diagnostic Properties of Different Serological Methods for Syphilis Testing in Brazil"

_diagnostics, 2025, doi:10.3390/diagnostics15121448_

Round 1

Reviewer 1 Report

Comments and Suggestions for Authors

The work is well written and proposes a diagnostic pathway for patients at risk for syphilis. The proposed pathway raises questions about some internationally recognized algorithms for Sexually Transmitted Diseases pathways. Generally, the work appears to be an application of a flow chart defined at the national level. While the manuscript addresses various realities and methods of diagnosis, it does not add significantly to existing literature. 
To enhance the discussion, it is recommended to include comments on the costs/benefits of screening and specify the relevant categories. Additionally, if the result of the syphilis test is negative, the authors could discuss the need for continued follow-up based on risk factors. In summary, the manuscript should extend beyond its microbiological-diagnostic value, as approved by the Ministry of Health or local authorities for sexually transmitted disease prevention, and incorporate broader perspectives.
Additionally, the introduction could be enhanced by providing information on risk groups and detailing the associated costs and consequences for women. 

Author Response

Diagnostic properties of different serological methods for syphilis testing in Brazil

REVIEWER 1

  1. Summary

Thank you very much for taking the time to review this manuscript. Please find the detailed responses below. The corresponding revisions and corrections have been highlighted in red in the resubmitted files. We are submitting an improved version of the manuscript, which we believe will be of interest to both researchers and clinicians.

  1. Questions for General Evaluation

Does the introduction provide sufficient background and include all relevant references?

Reviewer’s Evaluation: Can be improved

Are all the cited references relevant to the research?

Reviewer’s Evaluation: Can be improved

Is the research design appropriate?

Reviewer’s Evaluation: Can be improved

Are the methods adequately described?

Reviewer’s Evaluation: Can be improved

Are the results clearly presented?

Reviewer’s Evaluation: Can be improved

Are the conclusions supported by the results?

Reviewer’s Evaluation: Can be improved

  1. Point-by-point response to Comments and Suggestions for Authors

Comments 1: The work is well written and proposes a diagnostic pathway for patients at risk for syphilis. The proposed pathway raises questions about some internationally recognized algorithms for Sexually Transmitted Diseases pathways. Generally, the work appears to be an application of a flow chart defined at the national level. While the manuscript addresses various realities and methods of diagnosis, it does not add significantly to existing literature. 
To enhance the discussion, it is recommended to include comments on the costs/benefits of screening and specify the relevant categories. Additionally, if the result of the syphilis test is negative, the authors could discuss the need for continued follow-up based on risk factors. In summary, the manuscript should extend beyond its microbiological-diagnostic value, as approved by the Ministry of Health or local authorities for sexually transmitted disease prevention, and incorporate broader perspectives. Additionally, the introduction could be enhanced by providing information on risk groups and detailing the associated costs and consequences for women.

Response 1: We appreciate your comments. However, the main objective of this study was to assess the diagnostic properties of different tests for syphilis diagnosis and to compare the VDRL test, which is widely used in Brazil, with the RPR test, which is recommended by the WHO and used in most other countries. We observed that RPR titers tend to be lower compared to VDRL, a discrepancy that may impact clinical follow-up, particularly when monitoring low titer levels. To our knowledge, this is the first time this difference has been documented. Unfortunately, we do not have cost-effectiveness data; however, we agree that this would be an important topic for future research. Regarding risk factors for syphilis positivity, sex was not associated with a positive POC test. Given the high prevalence of syphilis in Brazil, we believe that screening efforts should target the general population, regardless of ethnicity or educational level, as these factors are associated with higher prevalence rates.

  1. Response to Comments on the Quality of English Language

Point 1: The English is fine and does not require any improvement.

Response 1: Thank you for your review.

Reviewer 2 Report

Comments and Suggestions for Authors

The manuscript titled “Diagnostic properties of different serological methods for syphilis testing in Brazil” by Wendland et al. has been thoroughly reviewed. The manuscript examines the variations in diagnostic modalities for syphilis detection, utilising positive point-of-care testing as the primary screening method for 125 samples.

Abstract:

The methodology and findings require further elucidation.  Why is RPR utilised as a reference standard for a non-treponemal test such as VDRL?  Both tests assess reagin antibodies and exhibit low sensitivity and specificity.  The results indicate that non-treponemal tests are compared to TPHA as the reference standard.  The context is quite perplexing.

Introduction:

The background is robust; however, it would benefit from the inclusion of information regarding the various phases of syphilis and how the tests can aid in diagnosis.

“However, no test achieves 100% sensitivity and specificity. False-negative, false-positive, indeterminate, or discrepant results between different tests are common in routine clinical laboratory practice.” This appears to be of mediocre quality; it would be beneficial if the data were based on local studies.

Bacterial names should be italicised.

Material and methods;

Please define “individuals with and without syphilis infection”

Kindly outline the criteria for inclusion and exclusion in the selection of samples from the subjects.

 Please provide a rationale for the use of POC in screening procedures.  The methodology for blinding samples in non-treponemal and treponemal tests is not outlined.  How is observer variability addressed in the interpretation of RPR and VDRL results?  The approach of conducting POC tests at the outset of the selection process, and incorporating them into assessments of sensitivity, specificity, and predictability, is challenging to accept.  The clinical parameters for assessing various phases of syphilis are insufficient.  Kindly provide a justification and discuss the implications for diagnostic performance.  What is the rationale behind the authors' decision not to stratify test performance according to disease stage?

 Each test's description must include detailed information regarding the specific antigens or antibodies utilised for POC and ELISA, as well as the diagnostic performance of the associated kits.

 What test serves as the reference standard?  What is the rationale behind the authors employing two distinct standards for these tests?

Results:

Kindly enhance Figure 3.  Kindly elucidate the rationale behind the presence of four algorithms.  And how do you discuss this?

Discussion

Kindly elaborate on the potential implications of the findings on the algorithm.

 Kindly elaborate on the types of antigens that may affect the results and ways in which this might be addressed.

 The discrepancies in the results, including false positives and negatives, as well as predictive values (Table 4), require further discussion and citation.

 Kindly provide a detailed explanation of the limitations.  Due to the limited sample size and the use of convenience sampling, the results cannot be broadly generalised.

 I believe chemiluminescence does not warrant lengthy elucidation.

 The text would benefit from an English proofreading service.

Comments on the Quality of English Language

 The text would benefit from an English proofreading service.

Author Response

Diagnostic properties of different serological methods for syphilis testing in Brazil

REVIEWER 2

  1. Summary

Thank you very much for taking the time to review this manuscript. Please find the detailed responses below. The corresponding revisions and corrections have been highlighted in red in the resubmitted files. We are submitting an improved version of the manuscript, which we believe will be of interest to both researchers and clinicians.

  1. Questions for General Evaluation

Is the work a significant contribution to the field?

Reviewer’s Evaluation: Must be improved

Is the work well organized and comprehensively described?

Reviewer’s Evaluation: Must be improved

Is the work scientifically sound and not misleading?

Reviewer’s Evaluation: Must be improved

Are there appropriate and adequate references to related and previous work?

Reviewer’s Evaluation: Must be improved

Is the English used correct and readable?

Reviewer’s Evaluation: Must be improved

  1. Point-by-point response to Comments and Suggestions for Authors

Comments 1: The manuscript titled “Diagnostic properties of different serological methods for syphilis testing in Brazil” by Wendland et al. has been thoroughly reviewed. The manuscript examines the variations in diagnostic modalities for syphilis detection, utilising positive point-of-care testing as the primary screening method for 125 samples.

Abstract:

The methodology and findings require further elucidation. Why is RPR utilised as a reference standard for a non-treponemal test such as VDRL?  Both tests assess reagin antibodies and exhibit low sensitivity and specificity.  The results indicate that non-treponemal tests are compared to TPHA as the reference standard.  The context is quite perplexing.

Response 1: We appreciate you bringing this to our attention. Although the RPR is widely used and recommended by the WHO, it is not currently utilized in Brazil. Therefore, conducting a comparative analysis of the diagnostic performance of both tests is essential. We also note that few studies have directly compared the sensitivity and specificity of RPR and VDRL. Additionally, we observed significant discrepancies in titers between the two tests, which is consistent with findings from previous studies. TPHA was used exclusively as the reference standard for treponemal tests. For clarity, we have revised the sentence in the abstract as follows: “TPHA was used as the reference standard for TT, and RPR as the reference standard for NTT.”

Comments 2: Introduction:

The background is robust; however, it would benefit from the inclusion of information regarding the various phases of syphilis and how the tests can aid in diagnosis.

“However, no test achieves 100% sensitivity and specificity. False-negative, false-positive, indeterminate, or discrepant results between different tests are common in routine clinical laboratory practice.” This appears to be of mediocre quality; it would be beneficial if the data were based on local studies.

Response 2: Thank you for your kind comments. We have revised the text and added a paragraph describing the differences in the diagnostic properties of treponemal and non-treponemal test according to the stage of syphilis. This addition highlights the diagnostic challenges associated with syphilis. The revised text reads as follows: “The performance of different serological tests varies throughout the stages of syphilis. In the early stage (primary syphilis), TT is generally more sensitive than NTT. Relying on non-treponemal tests during early infection may result in misdiagnosis and delays in treatment. As the disease progresses to the secondary stage, the sensitivity of non-treponemal tests increases and may approach 100%. Diagnostic challenges may also arise depending on the testing algorithm employed, as some laboratories prioritize specific tests as primary diagnostic tools, particularly when clinical symptoms are unclear, as is often the case with syphilis.”

Comments 3: Bacterial names should be italicised.

Response 3: We agree and have revised the nomenclature accordingly.

Comments 4: Please define “individuals with and without syphilis infection”

Response 4: We appreciate you bringing this to our attention. We have modified the sentence to clarify that individual classification was based on the positivity of the POC test: “This is a cross-sectional study that evaluated individuals with and without syphilis infection based on the results of the POC test and who were enrolled in the SIM study.”

Comments 5: Kindly outline the criteria for inclusion and exclusion in the selection of samples from the subjects.

Response 5: Thank you for your comment. We have modified the sentence to clarify how participants were selected for this study. The revised sentence reads: “All participants who tested positive for syphilis on the POC test were invited to provide a blood sample via venipuncture, collected in a BD Vacutainer® SST® II Advance® tube with separating gel for serological testing. Samples were centrifuged at 2000 rcf for 10 minutes to obtain serums. We used a random sample from a large prevalence study (SIM Study), along with a consecutive sample of participants who tested negative for all STIs evaluated by POC testing, to minimize potential cross-reactivity.”

Comments 6: Please provide a rationale for the use of POC in screening procedures. 

Response 6: POC testing is recommended by the Brazilian Ministry of Health for screening due to its ease of sample collection and rapid results.

Comments 7: The methodology for blinding samples in non-treponemal and treponemal tests is not outlined. 

Response 7: This study was conducted without the use of blinding.

Comments 8: How is observer variability addressed in the interpretation of RPR and VDRL results? 

Response 8: To minimize bias due to observer variability, all observations were performed by the same researcher. We have added the following sentences for clarification:

2.4.1 section: “All results were interpreted by the same observer.”

2.4.2 section: “As result interpretation was observer-dependent, all results in this study were evaluated by the same observer to ensure consistency.”

Comments 9: The approach of conducting POC tests at the outset of the selection process, and incorporating them into assessments of sensitivity, specificity, and predictability, is challenging to accept.  The clinical parameters for assessing various phases of syphilis are insufficient. Kindly provide a justification and discuss the implications for diagnostic performance. What is the rationale behind the authors' decision not to stratify test performance according to disease stage?

Response 9: We agree with the reviewer that the disease stage can impact test performance, and we have already addressed this in a paragraph in the Introduction. Unfortunately, as our samples were collected through a mobile screening unit, clinical information regarding disease stage was not available. Nevertheless, this strategy of large-scale population screening is crucial for enhancing diagnosis and treatment coverage, thereby contributing to the interruption of the transmission chain.

Comments 10: Each test's description must include detailed information regarding the specific antigens or antibodies utilised for POC and ELISA, as well as the diagnostic performance of the associated kits. What test serves as the reference standard? What is the rationale behind the authors employing two distinct standards for these tests?

Response 10: We thank the reviewer for this important observation. We have revised the manuscript to include detailed information on the specific antigens and antibodies targeted by each of the tests evaluated. For the POC and ELISA tests, we now specify whether they detect treponemal or non-treponemal antibodies and the nature of the antigens used, according to manufacturer data.

Regarding the reference standards, we used TPHA as the reference standard for treponemal tests and RPR as the reference standard for non-treponemal tests. This approach was based on current practices in syphilis diagnosis, where no single gold standard exists, and testing often involves a combination of treponemal and non-treponemal assays. Using distinct reference standards allowed us to evaluate the performance of each test within its appropriate diagnostic category.

We have clarified this rationale and the corresponding methodological details in the revised Methods section of the manuscript.

The following paragraph was added to the 2.3.1: “According to the manufacturer, the C region contains a polyclonal anti-T. pallidum antibody, and the T region contains a recombinant T. pallidum antigen immobilized on the membrane. Although the specific recombinant antigen is not disclosed, it is designed to bind to treponemal antibodies present in the sample. The manufacturer reports a clinical sensitivity greater than 99.9% and a specificity of 99.8% for the detection of treponemal antibodies.”

The following paragraph was added to the 2.3.2: “The manufacturer reports a clinical sensitivity greater than 99.9% and a clinical specificity also exceeding 99.9% for this assay. TPHA was used as the reference standard to evaluate the ELISA’s performance, consistent with current diagnostic protocols for treponemal testing.”

Comments 11: Kindly enhance Figure 3. 

Response 11: In the figure 3 we show the different flowcharts algorithms. Could you please be more specific regarding to modification requested? We improved the figure´s scale to enhance clarity and visualization.

Comments 12: Kindly elucidate the rationale behind the presence of four algorithms. And how do you discuss this?

Response 12: In the figure e we are showing only the reverse algorithms but with a sequence of different tests. In A and B, the non-treponemal test was VDRL followed by the treponemal test TPHA (A) and Elisa (B). In C and D, the non-treponemal test was RPR followed by the treponemal test TPHA (C) and Elisa (D).

Comments 13: Kindly elaborate on the potential implications of the findings on the algorithm.

Response 13: We rewrite the text to include the implication of the findings of the different flowcharts. The text now reads: “While all flowcharts presented comparable results demonstrating high sensitivity and specificity, VDRL alone exhibited higher rates of positivity when compared to RPR. Furthermore, ELISA is recognized for its enhanced sensitivity and specificity for syphilis in contrast to non-treponemal tests (e.g., VDRL). This observation is consistent with results reported by other researchers [5]. Therefore, incorporating VDRL can potentially eliminate the requirement for a third diagnostic test, and the use of ELISA can mitigate the occurrence of inconclusive findings.”

Comments 14: Kindly elaborate on the types of antigens that may affect the results and ways in which this might be addressed.

Response 14: Thank you for your comment. We added the following sentence to the Discussion section: “This is partly because NTT detect antibodies against non-specific antigens, such as cardiolipin, lecithin, and cholesterol, which can also be produced in response to various non-syphilitic conditions. To address this limitation, reactive NTT results should always be confirmed with a specific TT, especially in populations with high prevalence of conditions associated with false positives. Additionally, clinical correlation and, when available, follow-up testing can help distinguish between false positives and true active infections.”

Comments 15: The discrepancies in the results, including false positives and negatives, as well as predictive values (Table 4), require further discussion and citation.

Response 15: We have only Table 1 and 2 in the article.

Comments 16: Kindly provide a detailed explanation of the limitations.

Response 16: Thank you for your comment. We added the following sentence to the limitation section: “This lack of detailed clinical data constitutes an important limitation of our study, as it restricts the ability to fully assess the diagnostic accuracy across different disease stages.”

Comments 17: Due to the limited sample size and the use of convenience sampling, the results cannot be broadly generalised.

Response 17: We agree and have added the following sentence to the Discussion section: “Another important point to highlight is that the study was conducted in a subsample from the southern region of Brazil, which limits the generalizability of the results to other populations.”

Comments 18: I believe chemiluminescence does not warrant lengthy elucidation.

Response 18: Thank you for your comment. The paragraph regarding chemiluminescence was revised to present the information in a more concise manner, as we believe it is important to include this point in the manuscript. The text now reads: “Chemiluminescence is rarely used for syphilis diagnosis in Brazil due to its higher cost compared to traditional tests, the need for advanced laboratory infrastructure not widely available in remote or underserved areas, and the continued reliance on well-established, cost-effective methods like VDRL, FTA-ABS, and TPHA, which are considered efficient for large-scale screening.”

Comments 19: The text would benefit from an English proofreading service.

Response 19: The English language in the manuscript has been revised to improve clarity and comprehension.

  1. Response to Comments on the Quality of English Language

Point 1: The English could be improved to more clearly express the research.

Response 1: We have revised the English throughout the manuscript to improve understanding and overall flow.

Reviewer 3 Report

Comments and Suggestions for Authors

Dear Authors,

Your paper entitled "Diagnostic properties of different serological methods for syphilis testing in Brazil" has been carefully reviewed,

The article is very important from a medical point of view, since it tests the potential diagnostic use of different serological tests for the diagnosis of syphilis. 

The article is well written in English, well designed and attractive for readers,

Kindly find below a list of my comments regarding the present work:

01- In the Abstract section, you are invited to put the full scientific name of the test followed by its abbreviation. Test such as VDRL, RPR, ELISA, and TPHA, should be placed after the full name.

02- In the Abstract section, Line 24, you are invited to replace "for TT NTT, respectively." by "for TT and NTT, respectively."

03- In the whole manuscript you are invited to put the bacterial name in italic that is should be applied on Treponema pallidum and its abbreviation T. pallidum.

04- At the end of the Introduction section, the aim of the present study appear weak and too short, try please to highlight more on the aim of the study.

05- In the Materials and Methods section, Line 84, You mentioned "among others", You are kindly invited to replace it by the others demographics collected from participants.

06- In Figure 3 the words are too small, please provide this figure with a larger scale.

Best Regards

Author Response

Diagnostic properties of different serological methods for syphilis testing in Brazil

REVIEWER 3

  1. Summary

Thank you very much for taking the time to review this manuscript. Please find the detailed responses below. The corresponding revisions and corrections have been highlighted in red in the resubmitted files. We are submitting an improved version of the manuscript, which we believe will be of interest to both researchers and clinicians.

  1. Questions for General Evaluation

Does the introduction provide sufficient background and include all relevant references?

Reviewer’s Evaluation: Yes

Are all the cited references relevant to the research?

Reviewer’s Evaluation: Yes

Is the research design appropriate?

Reviewer’s Evaluation: Yes

Are the methods adequately described?

Reviewer’s Evaluation: Yes

Are the results clearly presented?

Reviewer’s Evaluation: Yes

Are the conclusions supported by the results?

Reviewer’s Evaluation: Yes

  1. Point-by-point response to Comments and Suggestions for Authors

Comments 1: In the Abstract section, you are invited to put the full scientific name of the test followed by its abbreviation. Test such as VDRL, RPR, ELISA, and TPHA, should be placed after the full name.

Response 1: Thank you for your suggestion. We agree, and the full scientific names have been added.

Comments 2: In the Abstract section, Line 24, you are invited to replace "for TT NTT, respectively." by "for TT and NTT, respectively."

Response 2: Thank you for your comment. The sentence has been revised for clarity.

Comments 3: In the whole manuscript you are invited to put the bacterial name in italic that is should be applied on Treponema pallidum and its abbreviation T. pallidum.

Response 3: Thank you for your comment. The bacterial name Treponema pallidum and its abbreviation T. pallidum have been italicized throughout the manuscript as recommended.

Comments 4: At the end of the Introduction section, the aim of the present study appear weak and too short, try please to highlight more on the aim of the study.

Response 4: Thank you for your suggestion. The sentence has been revised and reads: “The aim of this study was to evaluate the diagnostic properties of different serological methods for syphilis diagnosis and to assess the different diagnostic flowcharts used in Brazil.” 

Comments 5: In the Materials and Methods section, Line 84, You mentioned "among others", You are kindly invited to replace it by the others demographics collected from participants.

Response 5: Thank you for your suggestion. We have replaced the term "among others" with the specific sociodemographic variables collected from participants, namely marital status, income, employment status, place of residence, and nationality.

Comments 6: In Figure 3 the words are too small, please provide this figure with a larger scale.

Response 6: Thank you for your suggestion. We improved the figure´s scale to enhance clarity and visualization.

  1. Response to Comments on the Quality of English Language

Point 1: The English is fine and does not require any improvement.

Response 1: Thank you for your review.